# The Novel Role of SOX2 as an Early Predictor of Cancer Risk in Patients with Laryngeal Precancerous Lesions

**DOI:** 10.3390/cancers11030286

**Published:** 2019-03-01

**Authors:** Rocío Granda-Díaz, Sofía T. Menéndez, Daniel Pedregal Mallo, Francisco Hermida-Prado, René Rodríguez, Laura Suárez-Fernández, Aitana Vallina, Mario Sánchez-Canteli, Aida Rodríguez, M. Soledad Fernández-García, Juan P. Rodrigo, Juana M. García-Pedrero

**Affiliations:** 1Department of Otolaryngology, Hospital Universitario Central de Asturias and Instituto de Investigación Sanitaria del Principado de Asturias, University of Oviedo, Avda. Roma, 33011 Oviedo, Spain; rocigd281@gmail.com (R.G.-D.); sofiatirados@gmail.com (S.T.M.); pedregal.dm@gmail.com (D.P.M.); franjhermida@gmail.com (F.H.-P.); renerg.finba@gmail.com (R.R.); laura_quillo@hotmail.com (L.S.-F.); mariosanchezcanteli@gmail.com (M.S.-C.); aidarp.finba@gmail.com (A.R.); 2Ciber de Cáncer, CIBERONC, Av. Monforte de Lemos, 3-5, 28029 Madrid, Spain; 3Department of Pathology, Hospital Universitario Central de Asturias, Instituto Universitario de Oncología del Principado de Asturias, University of Oviedo, Avda. Roma, 33011 Oviedo, Spain; alaicla@hotmail.es (A.V.); solefghdr@hotmail.com (M.S.F.-G.)

**Keywords:** cancer risk assessment, larynx, dysplasia, SOX2, immunohistochemistry, gene amplification

## Abstract

The *SOX2* gene located at 3q26 is frequently amplified and overexpressed in multiple cancers, including head and neck squamous cell carcinomas (HNSCC). The tumor-promoting activity and involvement of SOX2 in tumor progression has been extensively demonstrated, thereby emerging as a promising therapeutic target. However, the role of SOX2 in early stages of tumorigenesis and its possible contribution to malignant transformation remain unexplored. This study investigates for the first time SOX2 protein expression by immunohistochemistry and gene amplification by real-time PCR using a large series of 94 laryngeal precancerous lesions. Correlations with the histopathological classification and the risk of progression to invasive carcinoma were established. Nuclear SOX2 expression was frequently detected in 38 (40%) laryngeal dysplasias, whereas stromal cells and normal adjacent epithelia showed negative expression. *SOX2* gene amplification was detected in 18 (33%) of 55 laryngeal dysplasias. Univariate Cox analysis showed that *SOX2* gene amplification (*p* = 0.046) and protein expression (*p* < 0.001) but not histological grading (*p* = 0.432) were significantly associated with laryngeal cancer risk. In multivariate stepwise analysis including age, tobacco, histology, *SOX2* gene amplification and SOX2 expression, SOX2 expression (HR = 3.531, 95% CI 1.144 to 10.904; *p* = 0.028) was the only significant independent predictor of laryngeal cancer development. These findings underscore the relevant role of SOX2 in early tumorigenesis and a novel clinical application of SOX2 expression as independent predictor of laryngeal cancer risk in patients with precancerous lesions beyond current WHO histological grading. Therefore, targeting SOX2 could lead to effective strategies for both cancer prevention and treatment.

## 1. Introduction

The sex-determining region Y (SRY)-related high-mobility-group (HMG)-box family of transcription factors member SOX2 (Sex-determining region Y-box 2) plays a critical role during embryonic development and organogenesis, thereby showing a very restricted, and precisely regulated, spatial-temporal expression pattern [1,2]. Similar to other pluripotency-associated transcription factors such as NANOG (Nanog Homeobox) and OCT4 (Octamer-binding transcription factor 4, also known as POU5F1), SOX2 has been implicated in sustaining stemness of embryonic stem cells, reprogramming of adult somatic cells to a pluripotent stem cell state and also in multiple tumorigenic processes [3,4,5,6,7,8,9,10]. 

Amplification of the chromosomal region 3q26-27 is one of the most recurrent genetic alterations in head and neck squamous cell carcinomas (HNSCC) and other carcinomas, which has been associated with tumor progression and poor patient prognosis [11,12]. *SOX2* located at 3q26 has emerged as a candidate tumor driver gene within this locus [7,13,14,15]. *SOX2* amplification and overexpression has been implicated in many tumor types, but mostly in squamous carcinomas of various localizations (lung, esophagus, head and neck) [13,14,15,16,17,18,19].

The role of SOX2 in HNSCC progression and its impact on prognosis and disease outcome has been subject of intense investigation [7,11,20,21,22,23]. However, the role of SOX2 in the early stages of HNSCC tumorigenesis and its possible contribution to malignant transformation and acquisition of an invasive phenotype remains unexplored.

This study investigates for the first time SOX2 protein expression and gene amplification in the early stages of HNSCC tumorigenesis using a large series of 94 laryngeal precancerous lesions. Correlations with the risk of progression to invasive carcinoma and with the histopathological classification (current gold standard) were established. Our findings uncover the clinical application of SOX2 expression as an independent predictor of laryngeal cancer risk in patients with laryngeal precancerous lesions, showing superior predictive value to the current World Health Organization (WHO) histological classification. 

## 2. Materials and Method

### 2.1. Patients and Tissue Specimens

Surgical tissue specimens from patients who were diagnosed of laryngeal dysplasia at the Hospital Universitario Central de Asturias between 1996 and 2010 were retrospectively collected, in accordance with approved institutional review board guidelines. All experimental procedures were conducted in accordance to the Declaration of Helsinki and approved by Institutional Ethics Committee of the Hospital Universitario Central de Asturias, and by the Regional CEIC from Principado de Asturias (date of approval: 18 July 2013; approval number: 81/2013) for the project PI13/00259. Informed consent was obtained from all patients. Patients must meet the following criteria to be included in the study: (i) pathological diagnosis of laryngeal dysplasia; (ii) with lesions of the vocal folds (iii) no previous history of head and neck cancer; (iv) complete excisional biopsy of the lesion; (v) a minimum follow-up of five years (or until progression to malignancy occurred); and (vi) patients with a diagnosis of laryngeal dysplasia who developed cancer within the next six months were excluded from the study. A total of 94 patients who met these criteria were included in this study. All the patients were treated with macroscopically complete excisional biopsy of the lesion, either with CO_2_ laser or with cold instruments. Microscopically surgical margins were not assessed. No other treatments were administered. Follow up with the patients occurred every two months in the first six months after completing the treatment, every three months until the second year, and every six months thereafter. 

Representative tissue sections from the original biopsy material were obtained from archival, paraffin embedded blocks and the histological diagnosis and epithelial dysplasia grade was confirmed in all the cases by an experienced pathologist (MSFG). The sections selected for study also contained normal epithelia as internal controls. The premalignant lesions were classified into the categories of low-grade and high-grade dysplasia following the WHO classification (4th Edition) [24]. 

### 2.2. Gene Amplification Analysis

The protocol for DNA extraction from paraffin-embedded tissue sections has been described elsewhere [25]. DNA extracted from normal mucosa obtained from non-oncologic patients was used as calibrator sample. Gene amplification was evaluated by real-time PCR (Q-PCR) in an ABI PRISM 7500 Sequence detector (Applied Biosystems, Foster City, CA, USA) using Power SYBR Green PCR Master Mix and oligonucleotides with the following sequences: for the *SOX2* gene, Fw, 5′- CTCCGGGACATGATCAGC-3′ and Rv, 5′- CTGGGACATGTGAAGTCTGC-3′; and for the reference gene *COL7A1* (located at 3p21), Fw, 5′- ACCCAGTACCGCATCATTGTG-3′ and Rv, 5′- TCAGGCTGGAACTTCAGTGTGT-3′. Samples were analyzed in triplicates and template-free blanks were also included. 

The relative gene copy number for *SOX2* was calculated using the 2^-ΔΔCT^ method. Calibration curves for the reference gene (*COL7A1*) and the target gene (*SOX2*) showed parallel slopes and comparable amplification efficiency across the linear range (1.5–25 ng) (Appendix A). The ΔΔC_T_ represents the difference between ΔC_T_ of dysplasia - ΔC_T_ of normal mucosa, with ΔC_T_ being the average C_T_ for the target gene (*SOX2*) minus the average C_T_ for the reference gene (*COL7A1*). The optimal cut-off value for *SOX2* amplification was identified via a receiver operating characteristic (ROC) curve analysis, using progression to cancer as an end point, and patients were categorized into positive *SOX2* amplification (≥1.75) and negative amplification (<1.75) (Appendix A). 

### 2.3. Immunohistochemistry

The formalin-fixed, paraffin-embedded tissues were cut into 3-µm sections and dried on Flex IHC microscope slides (Dako, Glostrup, Denmark). The sections were deparaffinized with standard xylene and hydrated through graded alcohols into water. Antigen retrieval was performed using Envision Flex Target Retrieval solution, high pH (Dako). Staining was done at room temperature on an automatic staining workstation (Dako Autostainer Plus, Dako, Glostrup, Denmark) with Anti-SOX2 rabbit polyclonal antibody (Merck Millipore # AB5603) at 1:1000 dilution using the Dako EnVision Flex + Visualization System (Dako Autostainer). Counterstaining with hematoxylin was the final step. 

SOX2 staining was evaluated as the percentage of cells with nuclear staining in the dysplastic epithelium. The optimal cut-off value for SOX2 staining calculated by ROC analysis using progression to cancer as end-point was 12.5% (Appendix A). Scores were classified as negative or positive staining on the basis of values below or above the cut-off value of 12.5%.

### 2.4. Statistical Analysis

χ^2^ and Fisher’s exact tests were used for comparison between categorical variables. For time-to-event analysis, Kaplan-Meier curves were plotted. Differences between survival times were analyzed by the log-rank method. Cox proportional hazards models were utilized for univariate and multivariate analyses. The hazard ratios (HR) with 95% confidence interval (CI) and *p* values were reported. All tests were two-sided. *p* values of ≤ 0.05 were considered statistically significant. 

## 3. Results

### 3.1. Patient Characteristics

All patients were men, with a mean age of 65 years (range 36–86 years). All but two patients were active or old smokers. The mean tobacco consumption was 50 packs-year (range 15–150 packs-year). After the diagnosis, all the patients that were active smokers received smoking cessation advice; however, 14 of them continued smoking. Of the 94 patients included in the study, 14 (15%) lesions were classified as low-grade dysplasia, and 80 (85%) as high-grade dysplasia. During the follow-up period, 29 (31%) of 94 patients developed an invasive carcinoma at the biopsy site.

The mean time to cancer diagnosis in the cases that progressed was 27 months (range 8 to 66 months). No significant differences attributable to age were observed (*p* = 0.501) between the patients who developed cancer (mean, 64 years) and those who did not (mean, 64 years). The mean tobacco consumption for patients who developed an invasive carcinoma was 58 packs per year, compared to 53 packs per year for those who did not develop cancer (*p* = 0.819). 

### 3.2. SOX2 Protein Expression in Laryngeal Precancerous Lesions

SOX2 protein expression was evaluated by immunohistochemistry in 94 laryngeal dysplasias. Nuclear SOX2 expression was detected in 38 (40%) dysplasias, whereas stromal cells and normal adjacent epithelia showed negative expression (Figure 1). 

SOX2 protein expression significantly correlated with the histopathological classification: 2 (14%) of the 14 low-grade dysplasias, and 36 (45%) of the 80 high-grade dysplasias exhibited SOX2-positive expression (Fisher’s exact test *p =* 0.039). 

### 3.3. SOX2 Gene Amplification during Laryngeal Tumorigenesis

*SOX2* gene amplification was assessed by real-time PCR on a set of 55 laryngeal dysplasias, using DNA extracted from the same paraffin tissue blocks. *SOX2* gene amplification was detected in 18 (33%) of 55 laryngeal dysplasias, with relative copy numbers ranging from 2-fold to 9-fold (Figure 2A). *SOX2* gene amplification increased with the severity of the lesions: 1 (17%) of the 6 low-grade dysplasias and 17 (35%) of the 49 high-grade dysplasias showed *SOX2* gene amplification, although the differences did not reach statistical differences (Fisher’s exact test *p =* 0.651). 

When analyzing the correlation between *SOX2* gene amplification and protein expression, we found that gene amplification only partially lead to SOX2 protein expression (Figure 2B). Thus, even though 22 out of 55 dysplasias were negative for both SOX2 expression and gene amplification, 15 amplification-negative lesions also showed positive SOX2 expression, indicating that additional mechanisms should contribute to the frequent SOX2 expression detected in laryngeal tumorigenesis. 

### 3.4. Associations with Laryngeal Cancer Risk

There was no statistically significant correlation in this cohort between the WHO histopathological grade and the risk of progression to laryngeal cancer (*p* = 0.538; Table 1; log-rank *p* = 0.426, Figure 3A), although high-grade dysplasias showed a higher cancer risk (HR = 1.615, 95% CI 0.489 to 5.336; *p* = 0.432; Table 1). Even the previous histological classification based in the dysplasia grading (mild, moderate vs. severe dysplasia) did not predict significantly cancer risk (log-rank *p* = 0.438, Figure 3B).

In marked contrast, we found that *SOX2* gene amplification significantly correlated with an increased laryngeal cancer risk (log-rank *p* = 0.038; Figure 3C). Furthermore, SOX2 protein expression showed the most robust association with laryngeal cancer risk (log-rank *p* < 0.001; Figure 3D). Five years after the patients were diagnosed, 20 (53%) of the 38 patients with SOX2-positive expression developed laryngeal cancer, whereas only 9 (16%) of the 56 patients with SOX2-negative expression progressed to invasive carcinoma (*p* < 0.001; Table 1). 

Univariate Cox analysis showed that *SOX2* gene amplification and protein expression but not histological grading were significantly associated with laryngeal cancer risk (Table 2). In multivariate stepwise analysis including age, tobacco, histology, *SOX2* gene amplification, and SOX2 expression, SOX2 expression (HR = 3.531, 95% CI 1.144 to 10.904; *p* = 0.028) was the only significant independent predictor of laryngeal cancer development. 

## 4. Discussion

There is mounting evidence demonstrating the role of SOX2 in tumorigenesis, and its contribution to tumor progression has been extensively documented in multiple cancers [7,8,9,10,19,20,21,22,23]. Accordingly, SOX2 has emerged as a candidate driver gene responsible for the 3q26-associated tumor aggressiveness [11,12,13,14,15,16,17,18]. In HNSCC, SOX2 expression has been found to induce cancer stem cell (CSC)-like properties, including increased self-renewal, tumorigenic potential and chemoresistence [26,27,28,29]. Accordingly, various studies using large cohorts of HNSCC samples have demonstrated that SOX2 expression significantly correlated with tumor recurrence and poor prognosis [20,28,29]. Although it has also been reported that low expression of SOX2 was associated with reduced survival and poor clinical outcome [30,31]. 

This study further and significantly extends these data investigating for the first time SOX2 protein expression and gene amplification in early stages of laryngeal tumorigenesis to ascertain its role in malignant transformation. Our findings demonstrate that SOX2 protein expression and gene amplification are frequent events in laryngeal tumorigenesis and, more importantly, both emerge as clinically and biologically relevant features in laryngeal cancer development. Even though SOX2 protein expression and *SOX2* amplification were found to increase in high-grade dysplasias, SOX2 protein expression and gene amplification but not histological grading were significantly associated with progression to laryngeal cancer. In particular, patients carrying SOX2-expressing dysplastic lesions exhibited a significantly higher cancer incidence than those with negative expression. In other tumor types, it has been reported that SOX2 antagonizes signals promoting differentiation to maintain stemness in CSC subpopulations [32,33]. On this basis, we could speculate that the increased cancer risk in patients harboring SOX2-expressing lesions may reflect the presence of a larger proportion of cells presenting cancer stem-like properties. Supporting this hypothesis, the expression of other CSC markers such as NANOG or Podoplanin has also been found to increase in precancerous lesions and to associate with a higher risk of malignization [34,35,36]. Podoplanin was identified as a marker of tumor-initiating cells in squamous cell carcinomas [37]. Tumorigenicity and capability of recapitulating human squamous cell carcinomas are by definition properties of CSCs. Thus, it has been interpreted that premalignant lesions with podoplanin expression expanding beyond the basal cell layer may represent truly early neoplastic lesions, enriched in CSC and indeed lesions with such clonal expansion carry a higher risk of progression to invasive cancer.

Furthermore, this study uncovers SOX2 expression as a robust independent predictor of laryngeal cancer risk beyond histological evaluation. Histopathological diagnosis of squamous intraepithelial lesions remains the gold standard in clinical practice for cancer risk assessment and decision-making [38]. Quite remarkably, the new and recently established WHO classification as well as the previous dysplasia grading failed to show a significant role in assessing laryngeal cancer risk in this cohort. This emphasizes the still limited value of histological grading to predict outcome, which is certainly affected by inter- and intra-observer variability [39]. Additional objective and reliable markers are therefore needed to improve patient stratification and to more accurately identify those carrying lesions at higher risk of progression who will require the most intense treatment and follow-up [40]. Our results clearly demonstrate that SOX2 protein analysis may provide valuable additional information beyond histological features. Hence, immunohistochemical evaluation of SOX2 is proposed to be incorporated into the clinical practice as a complementary and relatively simple molecular test for cancer risk assessment and decision-making. Nevertheless, routine implementation of SOX2 expression as biomarker will require confirmation in large prospective studies, while also extending analysis to other subsites in the head and neck area. 

The present study also revealed important temporal and mechanistic information regarding the early occurrence and frequency of SOX2 expression and gene amplification in laryngeal tumorigenesis. Thus, while both alterations are frequently detected at early stages of tumorigenesis and their frequency increased with the grade of dysplasia, *SOX2* gene amplification occurred at a lower frequency and did not perfectly match with protein expression, indicating that additional mechanisms must be contributing to SOX2 expression, such as transcriptional or posttranscriptional regulatory mechanisms. In line with this, it has been demonstrated that various transcription factors frequently altered in HNSCC, such as OCT4, YAP1 (Yes-Associated Protein 1) or the hypoxic factor HIF1α may induce the expression of SOX2 at mRNA level [41,42]. On the other hand, it has also been reported that amplification of *PIK3CA* and other genes mapping at 3q26 do not necessarily lead to increased expression, indicating that further epigenetic events could be involved in the transcriptional control [43]. Hence, this may explain some cases harboring *SOX2*-positive amplification that showed negative SOX2 protein expression. These results are in agreement with the TCGA data obtained from 279 HNSCC patients [44] using the platform cBioPortal for Cancer Genomics (http://cbioportal.org/) [45] (Appendix A). It has been extensively demonstrated (both experimentally and in silico) that 3q26 amplicon harbors numerous genes found to be frequently and concomitantly co-amplified and overexpressed in multiple cancers [46,47]. Various genes have been highlighted as highly significant oncogenic drivers for HNSCC survival [46], including the four known driver genes *PIK3CA, PRKCI, SOX2* and *TP63*. *In silico* analysis of TCGA HNSCC data included in Appendix A further illustrates that co-amplification of these four genes occurs frequently in HNSCC, thus showing that 3q26 amplification in HNSCC is not restricted to the *SOX2* gene. In addition, cytogenetic analyses demonstrated that 3q26 amplification is an early event in HNSCC tumorigenesis, and *PIK3CA* amplification and expression has been detected in dysplasias and associated with progression to invasive carcinoma [43].

## 5. Conclusions

This study provides the first evidence demonstrating the clinical relevance of SOX2 expression and gene amplification in early stages of laryngeal tumorigenesis. Our findings also uncover the clinical application of SOX2 expression as an independent predictor of laryngeal cancer risk in patients with precancerous lesions beyond current WHO histological grading. Hence, targeting SOX2 expression/function may potentially lead to the development of effective molecular-targeted therapies for both cancer prevention and treatment. Further investigation is therefore encouraged.

## Figures and Tables

**Figure 1 cancers-11-00286-f001:**
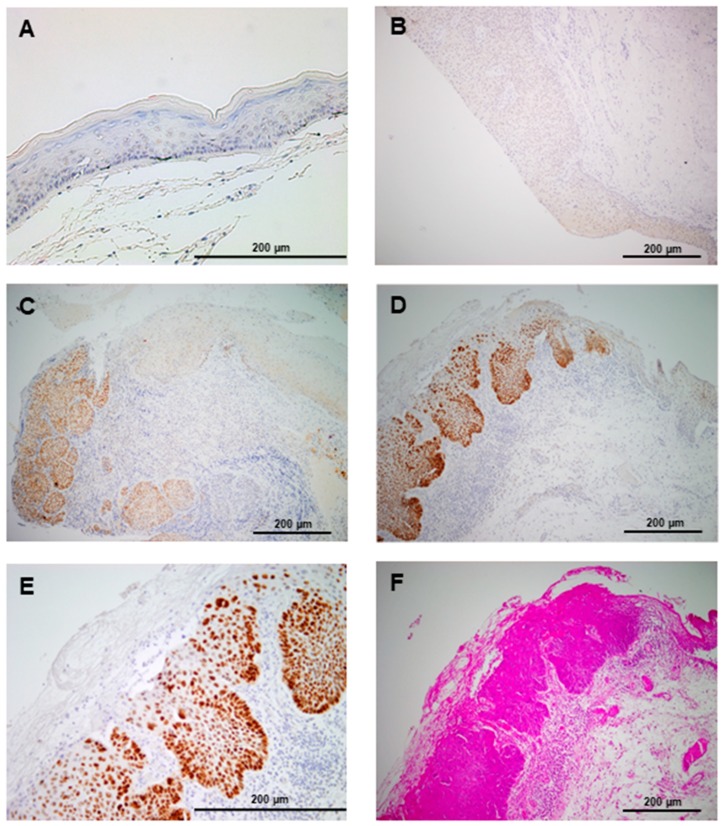
Immunohistochemical analysis of SOX2 (Sex-determining region Y-box 2) expression in laryngeal precancerous lesions. Normal adjacent epithelia showed negative staining (**A**). Representative examples of laryngeal dysplasias showing negative (**B**), and positive nuclear SOX2 staining (**C**,**D**), compared to the negative expression in normal-adjacent epithelia (right side). (**E**) Higher magnification from D. (**F**) H&E staining from D.

**Figure 2 cancers-11-00286-f002:**
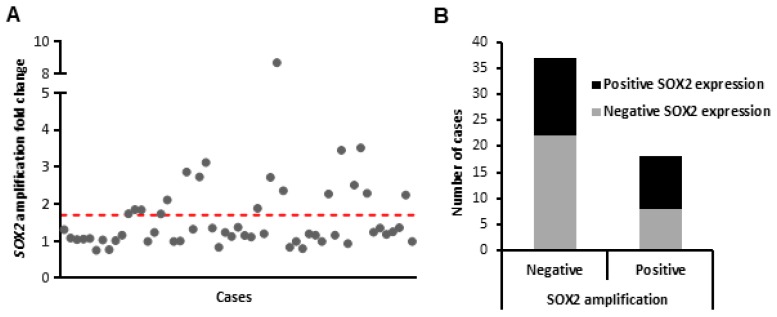
Analysis of *SOX2* gene amplification in laryngeal precancerous lesions. *SOX2* gene copy number was evaluated by Q-PCR in 55 cases. (**A**) Data are represented as fold-change in the precancerous lesion relative to the normal mucosa. Red dotted line indicates the threshold established to define positive cases (1.75). (**B**) Correlations between *SOX2* gene amplification and protein expression determined by immunohistochemistry.

**Figure 3 cancers-11-00286-f003:**
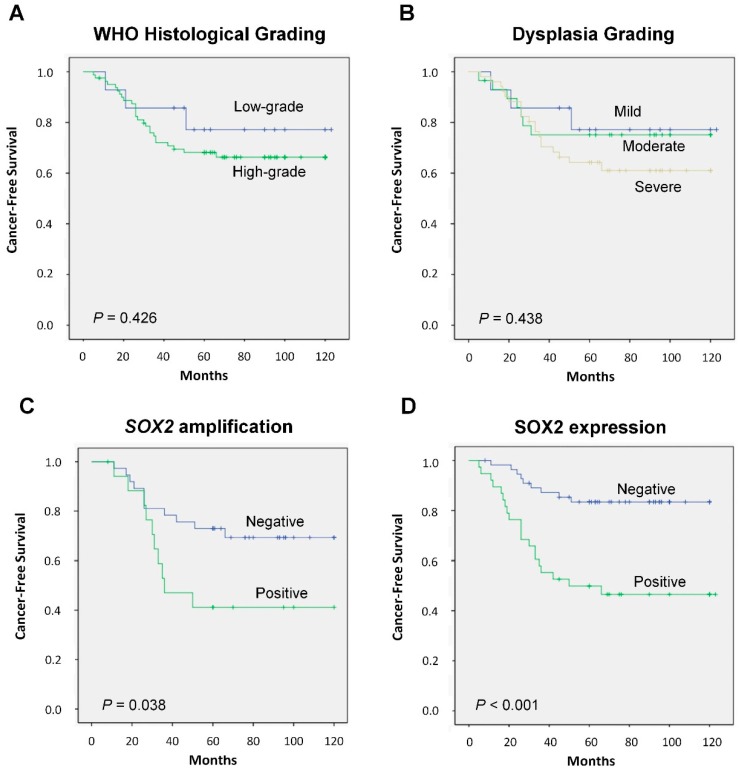
Kaplan-Meier cancer-free survival curves in the cohort of 94 patients with laryngeal dysplasias categorized by WHO (World Health Organization) histological grading (**A**), dysplasia grading (**B**) *SOX2* gene amplification (**C**) or SOX2 protein expression (**D**). *p* values were estimated using the log-rank test.

**Table 1 cancers-11-00286-t001:** Evolution of the premalignant lesions in relation to histopathological diagnosis, *SOX2* gene amplification, and protein expression.

Characteristic	No of Cases (%)	Progression to Carcinoma (%)	*p* ^†^
Histopathological diagnosis			
Low-grade dysplasia	14 (15)	3 (21)	0.538
High-grade dysplasia	80 (85)	26 (32)	
*SOX2* gene amplification			
Negative	37 (67)	11 (30)	0.081
Positive	18 (33)	10 (56)	
SOX2 protein expression			
Negative	56 (60)	9 (16)	<0.001
Positive (>10% stained nuclei)	38 (40)	20 (53)	

^†^ Fisher’s exact test.

**Table 2 cancers-11-00286-t002:** Univariate Cox Proportional Hazards Model to Estimate Laryngeal Cancer Risk.

Characteristic	*p*	Hazard Ratio	95% CI
Age (above vs. below the mean)	0.554	1.254	0.593–2.653
Smoking (above vs. below the mean)	0.618	1.210	0.572–2.558
Histology (high-grade vs. low-grade dysplasia)	0.432	1.615	0.489–5.336
*SOX2* amplification (positive vs. negative)	0.046	2.410	1.017–5.710
SOX2 expression (positive vs. negative)	<0.001	4.130	1.878–9.086

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
