# Peer review of "The Novel Role of SOX2 as an Early Predictor of Cancer Risk in Patients with Laryngeal Precancerous Lesions"

_cancers, 2019, doi:10.3390/cancers11030286_

Round 1

Reviewer 1 Report

The authors described the importance of SOX2 expression on laryngeal carcinogenesis. This finding is important for prediction of development in laryngeal dysplastic lesions.

However, I think there are some points which should be clarified this study is reliable or not.

1.    How do they determine SOX2 amplification positive or not? Do they perform all or none experiment? I think there should be cut off value to determine positive.

2.    They showed cut of value of SOX2 protein expression in immunohistochemistry. How do they determine this cut of value as 10%?

3.    In figure 3, positive or negative expression of SOX2 should be presented at the border of normal and dysplasia lesion to show SOX2 expression is specific in dysplastic lesion. Also, parallel HE staining of consecutive section should be present to identify normal or dysplasia clearly.

4.    Most importantly, the reason SOX2 gene negative and SOX2 protein positive should be clearly discussed. Because, they used paraffin embedded tissue for PCR, they should proof negative expression is not due to experimental failure. Usually, protein is more stable than gene for long time conservation. Their data showed stronger correlation in SOX2 protein than SOX2 gene, so this data is doubtful that it only reflect the stability of the target.

Author Response

Comments and Suggestions for Authors:

The authors described the importance of SOX2 expression on laryngeal carcinogenesis. This finding is important for prediction of development in laryngeal dysplastic lesions.

However, I think there are some points which should be clarified this study is reliable or not.

Point 1: How do they determine SOX2 amplification positive or not? Do they perform all or none experiment? I think there should be cut off value to determine positive.

Response 1: The analysis of SOX2 gene copy number was indeed performed in our cohort of patients with laryngeal dysplasia using Q-PCR, as described in Methods (Subsection 2.2) and Results (Subsection 3.3). As shown in Figure 2A, the cut-off value of 1.75 was used to determine positive gene amplification. In addition, in order to confirm our results on SOX2 amplification and to assess the correlation with SOX2 expression, we also performed in silico analysis of the publicly available TCGA data from 279 HNSCC patients (now Ref. 44) using the platform cBioPortal for Cancer Genomics (http://cbioportal.org/) (now Ref. 45). These data are shown in Supplementary Figure S2, and described in the Discussion (lines 247-249).

Point 2: They showed cut of value of SOX2 protein expression in immunohistochemistry. How do they determine this cut of value as 10%?

Response 2: The median value of 10% SOX2-positive nuclei was used as cut-off. This has now been clearly specified in Methods (lines 112-114) as follows: SOX2 staining was evaluated as the percentage of cells with nuclear staining in the dysplastic epithelium. SOX2 staining scores were classified as negative or positive staining on the basis of values below or above the median value of 10%. 

Point 3: In figure 3, positive or negative expression of SOX2 should be presented at the border of normal and dysplasia lesion to show SOX2 expression is specific in dysplastic lesion. Also, parallel HE staining of consecutive section should be present to identify normal or dysplasia clearly.

Response 3: The reviewer actually refers to Figure 1, which has now been modified according to his/her recommendation.

Point 4: Most importantly, the reason SOX2 gene negative and SOX2 protein positive should be clearly discussed. Because, they used paraffin embedded tissue for PCR, they should proof negative expression is not due to experimental failure. Usually, protein is more stable than gene for long time conservation. Their data showed stronger correlation in SOX2 protein than SOX2 gene, so this data is doubtful that it only reflect the stability of the target.

Response 4: Following the reviewer’s suggestion, we have further discussed (lines 240-247) some plausible explanations for the positive SOX2 protein expression observed in the absence of SOX2 gene amplification, such as transcriptional or posttranscriptional regulatory mechanisms. In line with this, it has been demonstrated that various transcription factors frequently altered in HNSCC, such as Oct4, YAP1 or the hypoxic factor HIF1α may induce the expression of SOX2 at mRNA level (new Refs. 41,42). On the other hand, it has also been reported that amplification of PI3KCA and other genes mapping at 3q26 not necessarily lead to increased expression, indicating that further epigenetic events could be involved in the transcriptional control (new Ref. 43). Hence, this may explain some cases harboring SOX2-positive amplification that showed negative SOX2 protein expression.

Regarding the stability of DNA obtained from paraffin-embedded tissue, this certainly was not a limitation for the analyses performed by Q-PCR, as for this purpose small DNA fragments for both SOX2 and the reference gene COL7A1 are amplified (<100 bp). In addition, the reference gene serves as a control of DNA quality. As such, the Ct values for COL7A1 gene in all the samples ranged between 25-27.

Reviewer 2 Report

This MS describes the detection of Sox2 amplification and/or increased expression in a large proportion of human laryngeal precancerous lesions, using surgical tissue specimens retroactively collected from 94 male patients. Sox2 Amplification and overexpression were frequent, but overexpression also occurred in the absence of gene amplification. Importantly, high Sox2 expression was a strong predictor for laryngeal cancer development.

The data presented are convincing and this study, although limited, should contribute to the assessment of the role of Sox2 in cancer.

Specific comments.

-the study does not suggest or provide any mechanism by which Sox2  would promote cancer development. It has been shown that in osteosarcomas high Sox2 expression marks and maintains cancer stem cells (CSC), (BasuRoy et al.Nature Comms 2016, Oncotarget 2016). Thus the bad prognosis for Sox2 expressing laryngeal lesions could be due to the presence of a large proportion of CSC in the original dysplastic lesions. This hypothesis (or others) should be discussed.

- although this maybe semantic, I am not sure of whether Sox2 should be classified as an oncogene. After all, ES cells express a lot of Sox2 and are not cancer cells. It could just be required for the maintenance and proliferation of CSC.

-line 112. Why a cut off of 10% positively stained cells? The lesions could be heterogeneous in cell composition, or contain relatively few CSC. Are the  cells neighboring the Sox2 positive cells really negative or low expressors?

- it would be interesting to know whether gene amplification is restricted to Sox2 or involves other neighboring genes.

Author Response

Comments and Suggestions for Authors:

This MS describes the detection of Sox2 amplification and/or increased expression in a large proportion of human laryngeal precancerous lesions, using surgical tissue specimens retroactively collected from 94 male patients. Sox2 Amplification and overexpression were frequent, but overexpression also occurred in the absence of gene amplification. Importantly, high Sox2 expression was a strong predictor for laryngeal cancer development.

The data presented are convincing and this study, although limited, should contribute to the assessment of the role of Sox2 in cancer.

Specific comments.

Point 1: the study does not suggest or provide any mechanism by which Sox2 would promote cancer development. It has been shown that in osteosarcomas high Sox2 expression marks and maintains cancer stem cells (CSC), (BasuRoy et al.Nature Comms 2016, Oncotarget 2016). Thus the bad prognosis for Sox2 expressing laryngeal lesions could be due to the presence of a large proportion of CSC in the original dysplastic lesions. This hypothesis (or others) should be discussed.

Response 1: We thank the reviewer for this insightful suggestion. Hypotheses on the mechanisms by which SOX2 could contribute to cancer development have now been discussed (lines 208-219), as follows:

In other tumor types, it has been reported that SOX2 antagonizes signals promoting differentiation to maintain stemness in CSC subpopulations [32,33]. On this basis, we could speculate that the increased cancer risk in patients harboring SOX2-expressing lesions may reflect the presence of a larger proportion of cells presenting cancer stem-like properties. Supporting this hypothesis, the expression of other CSC markers such as NANOG or Podoplanin has also been found to increase in precancerous lesions and to associate with a higher risk of malignization [34-36]. Podoplanin was identified as a marker of tumor-initiating cells in squamous cell carcinomas [37]. Tumorigenicity and capability of recapitulating human SCC are by definition properties of CSCs. Thus, it has been interpreted that premalignant lesions with podoplanin expression expanding beyond the basal cell layer may represent truly early neoplastic lesions, enriched in CSC and indeed lesions with such clonal expansion carry a higher risk of progression to invasive cancer.

Point 2: although this maybe semantic, I am not sure of whether Sox2 should be classified as an oncogene. After all, ES cells express a lot of Sox2 and are not cancer cells. It could just be required for the maintenance and proliferation of CSC.

Response 2: We fully agree that this is just a semantic issue. Similarly, multiple genes whose expression/function is altered in cancers and exhibit a pro-tumorigenic role are defined as oncogenes, despite existing the physiologic version in normal cells (i.e. proto-oncogenes). Numerous articles refer to SOX2 gene using the term “oncogene” (please see refs. 14-15). Nevertheless, to avoid confusion this term has been omitted in the text.

Point 3: line 112. Why a cut off of 10% positively stained cells? The lesions could be heterogeneous in cell composition, or contain relatively few CSC. Are the cells neighboring the Sox2 positive cells really negative or low expressors?

Response 3: The median value of 10% SOX2-positive nuclei was used as cut-off. This has now been clearly specified in Methods (lines 112-114), as follows: SOX2 staining was evaluated as the percentage of cells with nuclear staining in the dysplastic epithelium. SOX2 staining scores were classified as negative or positive staining on the basis of values below or above the median value of 10%. 

We fully agree with the reviewer that the lesions may contain a low proportion of CSC. Taking this into consideration, we have carefully reviewed all SOX2 immunostainings again to re-analyze the scoring data using a lower cut-off (>1% positive nuclei); however, it did not make any difference, because all the SOX2-positive lesions extend beyond 10% of dysplastic cells.

Even though it was generally observed that the SOX2-positive cells extend along each dysplastic area (perhaps fitting with the concept of clonal expansion of CSCs…). However, there were also some cases neighboring SOX2-negative nuclei or lower expression/intensity. Some representative images are uploaded to illustrate this (as Supplementary Information only for review).

Point 4: it would be interesting to know whether gene amplification is restricted to Sox2 or involves other neighboring genes.

Response 4: Indeed, it has been extensively demonstrated (both experimentally and in silico) that 3q26 amplicon harbors numerous genes found to be frequently and concomitantly co-amplified and overexpressed in multiple cancers (new refs. 46,47). Various genes have been highlighted as highly significant oncogenic drivers for HNSCC survival (new ref. 46), including the four known driver genes PIK3CA, PRKCI, SOX2 and TP63. In silico analysis of TCGA HNSCC data has now been included in Supplementary Figure S3 to further illustrate that co-amplification of these four genes occurs frequently in HNSCC, thus showing that 3q26 amplification in HNSCC is not restricted to SOX2 gene. In addition, cytogenetic analyses demonstrated that 3q26 amplification is an early event in HNSCC tumorigenesis and PIK3CA amplification and expression has been detected in dysplasias and associated with progression to invasive carcinoma (new ref. 43). This information has been included at the end of Discussion (lines 249-258).

Round 2

Reviewer 1 Report

Point 1: How do they determine SOX2 amplification positive or not? Do they perform all or none experiment? I think there should be cut off value to determine positive.

Response 1: The analysis of SOX2 gene copy number was indeed performed in our cohort of patients with laryngeal dysplasia using Q-PCR, as described in Methods (Subsection 2.2) and Results (Subsection 3.3). As shown in Figure 2A, the cut-off value of 1.75 was used to determine positive gene amplification. In addition, in order to confirm our results on SOX2 amplification and to assess the correlation with SOX2 expression, we also performed in silico analysis of the publicly available TCGA data from 279 HNSCC patients (now Ref. 44) using the platform cBioPortal for Cancer Genomics (http://cbioportal.org/) (now Ref. 45). These data are shown in Supplementary Figure S2, and described in the Discussion (lines 247-249).

Comments: Cut off  value should be determined by ROC analysis. Please show the ROC curve and data in detail.

Point 2: They showed cut of value of SOX2 protein expression in immunohistochemistry. How do they determine this cut of value as 10%?

Response 2: The median value of 10% SOX2-positive nuclei was used as cut-off. This has now been clearly specified in Methods (lines 112-114) as follows: SOX2 staining was evaluated as the percentage of cells with nuclear staining in the dysplastic epithelium. SOX2 staining scores were classified as negative or positive staining on the basis of values below or above the median value of 10%. 

Comments: Of course  I understand the method of evaluation. But, cut off  value should be determined by ROC analysis. Please show the ROC curve and data in detail.

Point 3: In figure 3, positive or negative expression of SOX2 should be presented at the border of normal and dysplasia lesion to show SOX2 expression is specific in dysplastic lesion. Also, parallel HE staining of consecutive section should be present to identify normal or dysplasia clearly.

Response 3: The reviewer actually refers to Figure 1, which has now been modified according to his/her recommendation.

Comments: That is fine.

Point 4: Most importantly, the reason SOX2 gene negative and SOX2 protein positive should be clearly discussed. Because, they used paraffin embedded tissue for PCR, they should proof negative expression is not due to experimental failure. Usually, protein is more stable than gene for long time conservation. Their data showed stronger correlation in SOX2 protein than SOX2 gene, so this data is doubtful that it only reflect the stability of the target.

Response 4: Following the reviewer’s suggestion, we have further discussed (lines 240-247) some plausible explanations for the positive SOX2 protein expression observed in the absence of SOX2 gene amplification, such as transcriptional or posttranscriptional regulatory mechanisms. In line with this, it has been demonstrated that various transcription factors frequently altered in HNSCC, such as Oct4, YAP1 or the hypoxic factor HIF1α may induce the expression of SOX2 at mRNA level (new Refs. 41,42). On the other hand, it has also been reported that amplification of PI3KCA and other genes mapping at 3q26 not necessarily lead to increased expression, indicating that further epigenetic events could be involved in the transcriptional control (new Ref. 43). Hence, this may explain some cases harboring SOX2-positive amplification that showed negative SOX2 protein expression.

Regarding the stability of DNA obtained from paraffin-embedded tissue, this certainly was not a limitation for the analyses performed by Q-PCR, as for this purpose small DNA fragments for both SOX2 and the reference gene COL7A1 are amplified (<100 bp). In addition, the reference gene serves as a control of DNA quality. As such, the Ct values for COL7A1 gene in all the samples ranged between 25-27.

Comments: That is fine. Thank you very much for considering.

Author Response

Comments and Suggestions for Authors:

Point 1: How do they determine SOX2 amplification positive or not? Do they perform all or none experiment? I think there should be cut off value to determine positive.

Response 1: The analysis of SOX2 gene copy number was indeed performed in our cohort of patients with laryngeal dysplasia using Q-PCR, as described in Methods (Subsection 2.2) and Results (Subsection 3.3). As shown in Figure 2A, the cut-off value of 1.75 was used to determine positive gene amplification. In addition, in order to confirm our results on SOX2 amplification and to assess the correlation with SOX2 expression, we also performed in silico analysis of the publicly available TCGA data from 279 HNSCC patients (now Ref. 44) using the platform cBioPortal for Cancer Genomics (http://cbioportal.org/) (now Ref. 45). These data are shown in Supplementary Figure S2, and described in the Discussion (lines 247-249).

Comments: Cut off value should be determined by ROC analysis. Please show the ROC curve and data in detail.

Response 1: Following the reviewer’s recommendation, the cut-off value for SOX2 amplification has been determined by ROC analysis. This information is shown in Supplementary Figure S2A and also described in the text (lines 102-105). Please note that the optimal cut-off calculated by ROC is 1.75, as we previously determined. Therefore, all the subsequent analyses and data in the Tables and text remain unchanged.

Point 2: They showed cut of value of SOX2 protein expression in immunohistochemistry. How do they determine this cut of value as 10%?

Response 2: The median value of 10% SOX2-positive nuclei was used as cut-off. This has now been clearly specified in Methods (lines 112-114) as follows: SOX2 staining was evaluated as the percentage of cells with nuclear staining in the dysplastic epithelium. SOX2 staining scores were classified as negative or positive staining on the basis of values below or above the median value of 10%. 

Comments: Of course I understand the method of evaluation. But, cut off value should be determined by ROC analysis. Please show the ROC curve and data in detail.

Response 2: The cut-off value for SOX2 expression has been determined by ROC analysis. This information is shown in Supplementary Figure S2B and also described in the text (lines 116-120). The optimal cut-off calculated by ROC curve analysis was 12.5%. Since there were no cases between 10-12.5%, all the SOX2 expression data remain unchanged.

Point 3: In figure 3, positive or negative expression of SOX2 should be presented at the border of normal and dysplasia lesion to show SOX2 expression is specific in dysplastic lesion. Also, parallel HE staining of consecutive section should be present to identify normal or dysplasia clearly.

Response 3: The reviewer actually refers to Figure 1, which has now been modified according to his/her recommendation.

Comments: That is fine.

Response 3: We thank the reviewer for considering that this was satisfactorily addressed.

Point 4: Most importantly, the reason SOX2 gene negative and SOX2 protein positive should be clearly discussed. Because, they used paraffin embedded tissue for PCR, they should proof negative expression is not due to experimental failure. Usually, protein is more stable than gene for long time conservation. Their data showed stronger correlation in SOX2 protein than SOX2 gene, so this data is doubtful that it only reflect the stability of the target.

Response 4: Following the reviewer’s suggestion, we have further discussed (lines 240-247) some plausible explanations for the positive SOX2 protein expression observed in the absence of SOX2 gene amplification, such as transcriptional or posttranscriptional regulatory mechanisms. In line with this, it has been demonstrated that various transcription factors frequently altered in HNSCC, such as Oct4, YAP1 or the hypoxic factor HIF1α may induce the expression of SOX2 at mRNA level (new Refs. 41,42). On the other hand, it has also been reported that amplification of PI3KCA and other genes mapping at 3q26 not necessarily lead to increased expression, indicating that further epigenetic events could be involved in the transcriptional control (new Ref. 43). Hence, this may explain some cases harboring SOX2-positive amplification that showed negative SOX2 protein expression.

Regarding the stability of DNA obtained from paraffin-embedded tissue, this certainly was not a limitation for the analyses performed by Q-PCR, as for this purpose small DNA fragments for both SOX2 and the reference gene COL7A1 are amplified (<100 bp). In addition, the reference gene serves as a control of DNA quality. As such, the Ct values for COL7A1 gene in all the samples ranged between 25-27.

Comments: That is fine. Thank you very much for considering.

Response 4: We also thank the reviewer for his/her insightful suggestion.

Round 3

Reviewer 1 Report

The authors adressed the all requests well.